# *Kras* Gene Analysis Using Liquid-Based Cytology Specimens Predicts Therapeutic Responses and Prognosis in Patients with Pancreatic Cancer

**DOI:** 10.3390/cancers14030551

**Published:** 2022-01-22

**Authors:** Masahiro Itonaga, Reiko Ashida, Shin-Ichi Murata, Yasunobu Yamashita, Keiichi Hatamaru, Takashi Tamura, Yuki Kawaji, Yuudai Kayama, Tomoya Emori, Manabu Kawai, Hiroki Yamaue, Ibu Matsuzaki, Hirokazu Nagai, Yuichi Kinoshita, Ke Wan, Toshio Shimokawa, Masayuki Kitano

**Affiliations:** 1Second Department of Internal Medicine, Wakayama Medical University, Wakayama 641-0012, Japan; itonaga@wakayama-med.ac.jp (M.I.); yasunobu@wakayama-med.ac.jp (Y.Y.); papepo51@wakayama-med.ac.jp (K.H.); ttakashi@wakayama-med.ac.jp (T.T.); y-kawaji@wakayama-med.ac.jp (Y.K.); kayama@wakayama-med.ac.jp (Y.K.); t-emori@wakayama-med.ac.jp (T.E.); kitano@wakayama-med.ac.jp (M.K.); 2Second Department of Surgery, School of Medicine, Wakayama Medical University, Wakayama 641-0012, Japan; smurata@wakayama-med.ac.jp (S.-I.M.); kawai@wakayama-med.ac.jp (M.K.); yamaue-h@wakayama-med.ac.jp (H.Y.); 3Department of Human Pathology, Wakayama Medical University, Wakayama 641-0012, Japan; m_ibu@wakayama-med.ac.jp (I.M.); nagnag0896@gmail.com (H.N.); y.kinoshita.wakayama@gmail.com (Y.K.); 4Clinical Study Support Center, Wakayama Medical University, Wakayama 641-0012, Japan; kwan@wakayama-med.ac.jp (K.W.); shimokaw@wakayama-med.ac.jp (T.S.)

**Keywords:** *Kras*, pancreatic ductal adenocarcinoma, pancreatic cancer, gemcitabine and nab-paclitaxel, EUS-FNA, liquid-based cytology

## Abstract

**Simple Summary:**

New therapeutic strategies are needed to improve the prognosis of pancreatic ductal adenocarcinoma (PDAC) and developing biomarkers that can guide individualized treatment decisions is an important part of these strategies. In this study, we found that unresectable PDAC patients harboring wild-type *Kras* had significantly longer progression-free survival (PFS) and overall survival (OS) than those harboring mutant *Kras* after undergoing first-line gemcitabine and nab-paclitaxel (GA) therapy and that wild-type *Kras* was a significant predictor of longer PFS and OS. This is the first report suggesting that *Kras* gene analysis has the potential to predict therapeutic responses to GA and the prognosis of unresectable PDAC.

**Abstract:**

Background: Although several molecular analyses have shown that the *Kras* gene status is related to long-term survival of patients with pancreatic ductal adenocarcinoma (PDAC), the results remain controversial. Here, we examined the *Kras* gene status in a cohort of unresectable PDAC patients who underwent first-line therapy with gemcitabine and nab-paclitaxel (GA) and assessed differences in chemotherapy responses and survival. Methods: Patients with a histological diagnosis of PDAC (based on EUS-guided fine-needle aspiration) from 2017 to 2019 were enrolled. Tumor genomic DNA was extracted from residual liquid-based cytology specimens and *Kras* mutations were assessed using the quenching probe method. The relationships between the *Kras* status and progression-free survival (PFS) and overall survival (OS) were assessed. Results: Of the 110 patients analyzed, 15 had wild-type *Kras*. Those with the wild-type gene showed significantly longer PFS and OS than those with mutant *Kras* (6.9/5.3 months (*p = 0.044*) vs. 19.9/11.8 months (*p = 0.037*), respectively). Multivariate analyses identified wild-type *Kras* as a significant independent factor associated with longer PFS and OS (HR = 0.53 (*p = 0.045*) and HR = 0.35 (*p = 0.007*), respectively). Conclusions: The analysis of the *Kras* gene status could be used to predict therapeutic responses to GA and prognosis in unresectable PDAC patients.

## 1. Introduction

Pancreatic ductal adenocarcinoma (PDAC) is the third leading cause of cancer-related death in the United States [1] and is projected to become the second leading cause of cancer-related death worldwide in the next decade [2]. At the time of diagnosis, most patients have locally advanced or metastatic disease and <20% of patients have curatively operable disease [3]. Chemotherapy is the mainstay of treatment for locally advanced and metastatic disease and novel chemotherapy regimens such as FOLFIRINOX and gemcitabine plus nab-paclitaxel (GA) have improved prognosis for these patients [4,5]. Although progress has been made, long-term survival rates remain low, with <10% of patients remaining alive 5 years after diagnosis [6]. New treatment strategies are needed to significantly improve survival. One of the most promising tools is to establish biomarkers that can guide individualized treatment decisions.

Liquid-based cytology (LBC) is a technique commonly used (alongside conventional smear (CS) preparation) for the analysis of specimens obtained by endoscopic ultrasound-guided fine-needle aspiration (EUS-FNA). LBC is a thin-layer slide preparation technique developed to overcome the disadvantages of CS, such as cell clouding and blood contamination [7,8,9]. Moreover, since most of the collected cells can be used as LBC samples, a genomic analysis can be performed easily using the LBC samples that remain after cytological diagnosis [10]. Sekita-Hatakeyama et al. described the usefulness of residual LBC specimens for *Kras* mutation analysis in pancreatic cancer [11].

Mutations in the *Kras* oncogene are driver mutations; as such, they are an initiating event in PDAC. These mutations are present in >90% of PanIN lesions and in >90% of PDAC cases [12]. A meta-analysis of the data from 17 retrospective studies reported that the detection of *Kras* mutations might be a useful poor prognostic factor for PDAC [13]. However, this meta-analysis included data from patients in different clinical settings and treatments. Moreover, a multicenter prospective study by Hass et al. reported no association between *Kras* mutations and prognosis [14]. Thus, it is still controversial whether *Kras* mutations contribute to the poor prognosis of PDAC. In addition, the association between *Kras* mutations and prognosis needs to be evaluated in cases with standardized background factors (such as clinical setting and treatment).

Here, we analyzed *Kras* mutations using residual LBC specimens from EUS-FNA of patients with unresectable PDAC who underwent first-line GA therapy. We then examined differences in response to chemotherapy and survival.

## 2. Materials and Methods

### 2.1. Study Design

The protocol for this single-center cohort study was approved by the Wakayama Medical University Ethical Review Board (no. 3253). The study was conducted in accordance with the tenets of the Declaration of Helsinki.

The primary endpoints were PFS and OS of patients with unresectable PDAC who underwent first-line GA chemotherapy. These endpoints were evaluated according to differences in the *Kras* mutation status in residual LBC specimens from EUS-FNA. The factors predictive of PFS and OS were also analyzed.

### 2.2. Patients

The study included LBC specimens from 278 consecutive patients who underwent EUS-FNA at Wakayama Medical University Hospital between February 2017 and December 2019. Written informed consent was obtained from all research participants.

Inclusion criteria were (1) age ≥ 20 years, (2) a pathological diagnosis of PDAC by EUS-FNA, (3) unresectable locally advanced or metastatic PDAC and (4) first-line treatment chemotherapy using a GA regimen. Exclusion criteria were (1) LBC cytology class I or class II, (2) unsuccessful *Kras* mutation measurement and (3) less than one full course of first-line GA chemotherapy.

### 2.3. EUS-FNA Procedure and Specimen Processing

Patients underwent EUS-FNA using a GF-UCT260 linear echoendoscope (Olympus Medical, Tokyo, Japan) connected to an ultrasound scanning system (ARIETTA 850; Fujifilm, Tokyo, Japan). The pancreatic lesion was punctured using a 22/25G aspiration needle (Expect^TM^ or Acquire^TM^; Boston Scientific, Natick, MA, USA; or EZ Shot3^TM^; Olympus Medical, Tokyo, Japan) under real-time ultrasound guidance. The stylet was withdrawn and aspirated (10 cc negative pressure) using the attached syringe and the aspiration needle was moved back and forth 20 times within the lesion before being withdrawn from the echoendoscope. Finally, the aspirated material was pushed out into a preservative liquid (ThinPrep CytoLyt Solution; Hologic, Marlborough, MA, USA) by reinsertion of the stylet.

The aspirated material was separated for histological evaluation, cytological evaluation and *Kras* mutation analysis. The solid materials were fixed in 10% formalin, embedded in paraffin and sliced thinly for additional immunostaining as required. The liquid material was treated using the ThinPrep^®^ method according to the manufacturer’s recommendations and then evaluated immediately by Papanicolaou staining. The residual LBC specimens were stored at 4 °C until DNA extraction.

### 2.4. DNA Extraction and Kras Mutation Analysis

Residual LBC samples were centrifuged at 2000 rpm for 10 min. Genomic DNA was purified from the sediment using Maxwell RSC FFPE Kit-PKK Custom (Promega, Madison, WI, USA). The amount of DNA quantity was measured using a Quantus™ Fluorometer (Promega) and a QuantiFluor ONE dsDNA System (Promega). Gene mutations were examined for *Kras* codons (codons 12, 13, 59 and 61) and for the corresponding wild-type *Kras* using the fully automated genotyping system i-densy (IS-5320; ARKRAY, Tokyo, Japan) [15,16].

### 2.5. Definitions

Tumor diameter on contrast-enhanced computed tomography (CE-CT) was measured before chemotherapy. Responses to chemotherapy were classified according to the RECIST guidelines (ver. 1.1) as follows: complete response (CR), partial response (PR), stable disease (SD), or progressive disease (PD). The tumor reduction rate was calculated from the CE-CT as follows: (tumor size before chemotherapy minus tumor size after chemotherapy)/tumor size before chemotherapy. PR was defined as a 30% decrease in the longest diameter and PD as a 20% increase in the longest diameter. Observations were performed once every 2–3 months during the course of treatment.

### 2.6. Chemotherapy

Almost all patients received nab-paclitaxel (125 mg/m^2^) and gemcitabine (1000 mg/m^2^) as first-line chemotherapy. In some cases, the dose was reduced according to the general condition of the patient, such as age and performance status. Patients were followed carefully after initial treatment (by imaging and monitoring of tumor markers). Patients were treated with GA therapy until PD was observed. Patients with PD were offered a second-line chemotherapy regimen or the best supportive care. The start date of the follow-up was set as the date of initiation of first-line chemotherapy for PDAC. The end date of the follow-up was set as the final follow-up in May 2021 or the time of patient death.

### 2.7. Statistical Methods

The primary endpoints were PFS and OS of patients with unresectable PDAC who underwent first-line GA chemotherapy. With respect to background data, the significance of the differences in continuous data was assessed using non-paired Student’s *t*-tests as a reference. The chi-squared test was used to analyze categorical data. PFS and OS were measured from the first day of chemotherapy to the date of PD and death, respectively. The survival curves for PFS and OS were estimated using the Kaplan–Meier method.

Univariate and multivariate analyses using a Cox proportional hazard model were performed to identify variables significantly associated with PFS and OS. *p*-values < 0.05 were considered statistically significant. The statistical analyses were performed using JMP Pro ver. 14 (SAS Institute, Inc., Cary, NC, USA).

## 3. Results

Figure 1 shows the study flow chart. During the study period, the Kras gene statuses of 278 consecutive patients were examined. Of the patients assessed for enrollment, 168 were excluded and 110 were analyzed. The demographic and baseline characteristics of these patients are summarized in Table 1. There were 95 patients with mutant *Kras* and 15 with wild-type *Kras*. A total of 69 patients (63%) received second-line treatment, such as TS-1 (*n* = 51), FOLFIRINOX (*n* = 11), or nanoliposomal irinotecan plus fluorouracil and folinic acid (*n* = 7). The other patients were provided with the best supportive care. Patient factors such as sex, age, ECOG performance status, lesion size, disease status, CA19-9 expression, amount of extracted DNA and receipt of second-line chemotherapy were not significantly different between the two groups. However, patients with wild-type *Kras* had more head lesions than patients with mutant *Kras* (*p* = 0.012), although there was no significant difference in tumor size (Table 2).

The comparisons of response to first-line chemotherapy with GEM and nab-PTX were assessed between the wild-type *Kras* and the mutant *Kras* groups. According to RECIST v1.1, in the wild-type *Kras* group, 3 (20%) showed a PR, 10 (66.7%) showed SD and 2 (13.3%) showed PD and, in the mutant *Kras* group, 15 patients (15.8%) showed a PR, 40 (42.1%) showed SD and 40 (42.1%) showed PD. The rate of objective response was observed in 3 patients (20%) in the wild-type *Kras* group and 15 patients (15.8%) in the mutant *Kras* group (*p* = 0.701; Table 3). The rate of disease control was observed in 13 patients (86.7%) in the wild-type *Kras* group and 55 patients (57.9%) in the mutant *Kras* group (*p* = 0.044; Table 3). There was a significant difference in the rate of disease control between two groups.

The PFS and OS of patients, classified according to the *Kras* status, are shown in Figure 2 and Figure 3. The median PFS in the wild-type *Kras* and the mutant *Kras* groups were 6.9 months (95% CI = 3.3–11.2) and 5.3 months (95% CI = 2.3–6.2), respectively, and the median OS was 19.9 months (95% CI = 11.1–NA) and 11.8 months (95% CI = 7.3–18.4), respectively. Both PFS and OS were significantly longer in the wild-type *Kras* group (*p* = 0.044 and *p* = 0.037, respectively; log-rank test; see Figure 2 and Figure 3).

Factors associated with PFS and OS were identified using the Cox proportional hazard model. Only wild-type *Kras* was associated significantly with longer PFS in the univariate analysis (HR = 0.56, 95% CI = 0.31–0.99; *p = 0.049*) and multivariate analysis (HR = 0.53, 95% CI = 0.28–0.99; *p = 0.045*) (Table 4). Local advanced disease (HR = 0.60, 95% CI = 0.37–0.96; *p* = 0.026); *Kras,* wild-type (HR = 0.50, 95% CI = 0.26–0.97; *p* = 0.026); and second-line chemotherapy (HR = 0.44, 95% CI = 0.29–0.68; *p* < 0.001) were significantly associated with longer OS in the univariate analysis (Table 5). the multivariate analysis identified local advanced disease (HR = 0.57, 95% CI = 0.36–0.92; *p* = 0.048); *Kras,* wild-type (HR = 0.35, 95% CI = 0.16–0.74; *p* = 0.007); and second-line chemotherapy (HR = 0.20, 95% CI = 0.20–0.50; *p* < 0.001) as significant independent factors associated with longer OS (Table 5).

## 4. Discussion

In this study, we analyzed *Kras* mutations in residual LBC specimens of EUS-FNA samples from patients with unresectable PDAC who underwent first-line GA therapy. We then assessed the differences in response to chemotherapy and survival. We found that patients with wild-type *Kras* had significantly longer PFS and OS than those with mutant *Kras*, suggesting that wild-type *Kras* could be used as a predictor of longer PFS and OS. These findings suggest that *Kras* gene analysis can predict therapeutic responses to GA, as well as the prognosis of patients with PDAC.

Previous studies attempted to determine the role of *Kras* as a prognostic biomarker for the clinical outcome of PDAC. A meta-analysis that pooled data from 17 retrospective studies reported that *Kras* mutations might be a useful prognostic factor for PDAC [13]. However, this meta-analysis had two major problems. First, some cohorts included fewer cases with the *Kras* mutation (60–70% of cases) than other studies, which usually report 90% prevalence of *Kras* mutations, suggesting that the gene analyses used in these studies were of low quality. Second, the analysis included cases from various clinical settings with different treatments, such as surgery, chemotherapy and the best supportive care, in which the patients’ backgrounds were also not consistent. Haas et al. conducted a multicenter prospective study of unresectable PDAC and reported no significant difference in OS between patients with wild-type Kras and mutant Kras (9.9 vs. 8.3 months, *p* = 0.70, respectively) PDAC [14]. By contrast, an observational study of unresectable PDAC by Windon et al. noted that the OS of wild-type *Kras* patients was higher than that of mutant *Kras* patients (736 vs. 420 days, *p* = 0.026, respectively) [17]. This discrepancy may be due to differences in first-line chemotherapy regimens among patients. Therefore, in the present study, first-line chemotherapy was standardized to reduce bias related to patient background.

Although the mechanism for the longer PFS and OS of wild-type *Kras* patients compared with mutant *Kras* patients after first-line GA therapy was not elucidated, we hypothesized as follows. First, mutations in *Kras* cause constitutive activation of *RAS* and perpetuate downstream signaling pathways involved in cellular proliferation, migration, apoptosis and cytoskeletal remodeling, independent of growth factor receptor activation [18].

Sustained activation of *RAS* signaling had been reported as a major cause of gemcitabine resistance [19] and inhibiting *Kras* signaling suppresses migration and invasion in a gemcitabine-resistant cell line [20,21]. Therefore, *Kras* gene mutation may contribute to acquiring resistance and poor response to gemcitabine. Second, Schlitter et al. performed a histological analysis of surgical resection specimens and found that patients with colloid, medullary, or papillary carcinoma survived significantly longer than patients with conventional PDAC (*p =* 0.04) [22]. These findings suggest that wild-type *Kras* PDACs may be less likely to grow and spread rapidly than *Kras*-mutant PDACs. Third, on the basis of genome profile analyses, it was report that, compared with mutant *Kras* PDACs, wild-type *Kras* PDACs have one of the following three characteristics: (1) presence of an activated-MAPK in the absence of a *Kras* mutation, such as deleterious genetic changes in BRAF, GNAS, EGFR, ERBB2, MET, ERBB3, MAP2K4, FGFR1, NTRK1 and ERBB4; (2) presence of microsatellite instability/defective DNA mismatch repair; or (3) presence of kinase-fusion genes, such as FGFR2, RAF, ALK, RET, MET, NTRK, ERBB4 and FGFR3 [23,24,25]. These gene mutations may contribute to a better response to GA, although we did not analyze theses gene mutations in this study and the mechanism is unknown. In the future, it will be necessary to analyze not only the *Kras* mutation status but also the whole genome mutational landscape including these mutations to examine the response to GA and OS.

As for *Kras* gene analysis, formalin-fixed paraffin-embedded (FFPE) tissues or frozen tissue samples are commonly used for the genomic analysis of PDAC [26]. However, the amount of specimen obtained by EUS-FNA of a pancreatic lesion can be minimal; therefore, it is not always possible to obtain enough DNA for a multi-gene analysis. By contrast, the advantage of the molecular analysis of LBC specimens is that it can be performed using only liquid specimens, meaning that the remaining specimens used for cytology can be used [27]. Several reports described *Kras* mutation analysis using LBC specimens from lesions in other organs, especially the lungs. Zhao et al. reported that they successfully extracted DNA from LBC samples collected from EUS-FNA tumor tissues of lung cancer patients and performed real-time PCR in all cases [28]. In our study, we successfully detected *Kras* mutations in 98.6% (274/278 cases) of residual LBC specimens, suggesting the usefulness of the genomic analysis using LBC specimens. Furthermore, even if a sufficient amount of tissue can be obtained by EUS-FNA, it needs to be stored for future MSI testing or multi-gene panel analysis. Therefore, it is advantageous to use LBC specimens rather than FFPE for *Kras* single-gene analysis at the time of diagnosis [29,30].

The study has several limitations. First, it is a single-center retrospective study enrolling a relatively small number of patients to standardize patient backgrounds. In the future, a large-scale, multicenter study will be necessary. Second, the Tm analysis method using a quenching probe and i-densy^TM^ may enable a complete and fully automated analysis of polymorphisms in extracted DNA within about 60 min; further, the detection of *Kras* mutations using this method is better than the direct sequencing method (indeed, it is equivalent to the Scorpion-ARMS method, which has relatively high sensitivity) [31]. However, the major problem with this method is that it cannot detect *Kras* mutation subtypes. Currently, therapeutic drugs targeting *Kras* G12C are being developed and it will be necessary to detect *Kras* subtypes in the future [32]. Third, the presence and type of second chemotherapy and disease status (local or metastatic) were not standardized in our study. The second chemotherapy and disease status are considered to be very important for OS. Although the *Kras* status, presence of second chemotherapy and disease status were found to be independent factors for OS, further studies are needed to investigate the relationship between the *Kras* status and OS in patients who have received the same chemotherapy regimens and have the same disease status.

## 5. Conclusions

In conclusion, patients with PDAC harboring wild-type *Kras* had significantly longer PFS and OS than those with mutant *Kras*. In addition, the multivariate analysis revealed that wild-type *Kras* was a significant predictor of longer PFS and OS of patients with unresectable PDAC who underwent first-line GA therapy. These findings suggest that *Kras* gene analysis can predict therapeutic responses to GA, as well as the prognosis of patients with PDAC.

## Figures and Tables

**Figure 1 cancers-14-00551-f001:**
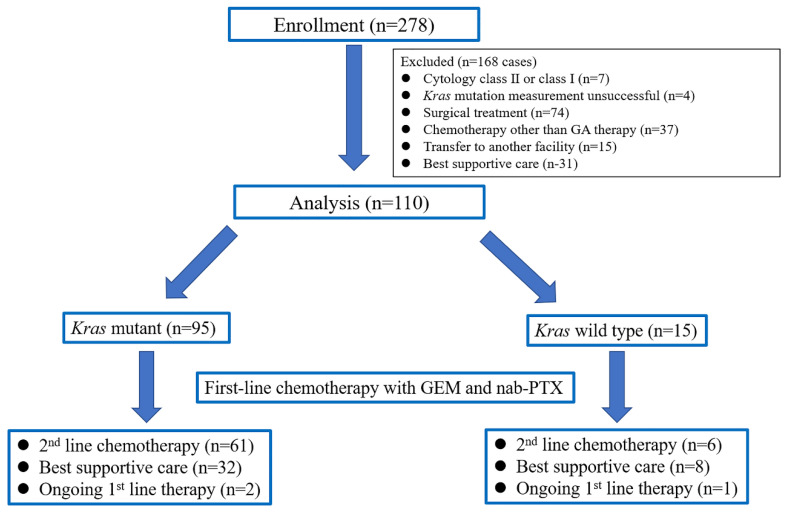
Study flow chart.

**Figure 2 cancers-14-00551-f002:**
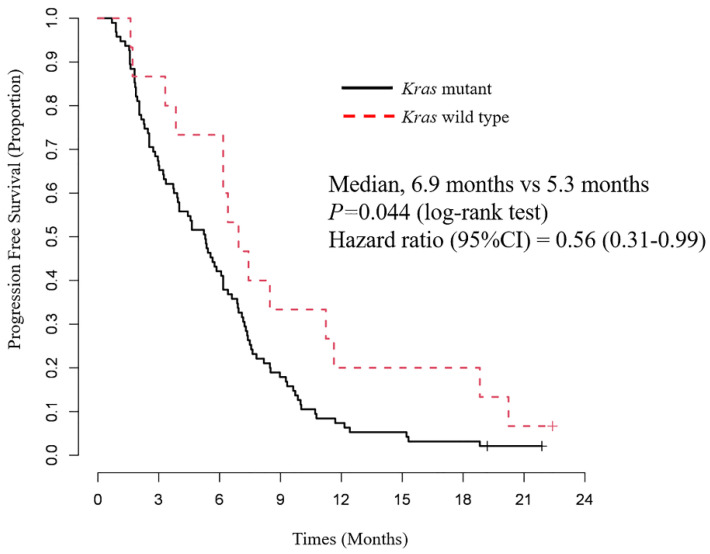
Progression–free survival according to *Kras* status after first–line chemotherapy with GEM and nab–PTX. The median PFS in the wild-type *Kras* and mutant *Kras* groups was 6.9 and 5.3 months, respectively. PFS in the wild-type *Kras* group was significantly longer than that in the mutant *Kras* group (*p* = 0.044).

**Figure 3 cancers-14-00551-f003:**
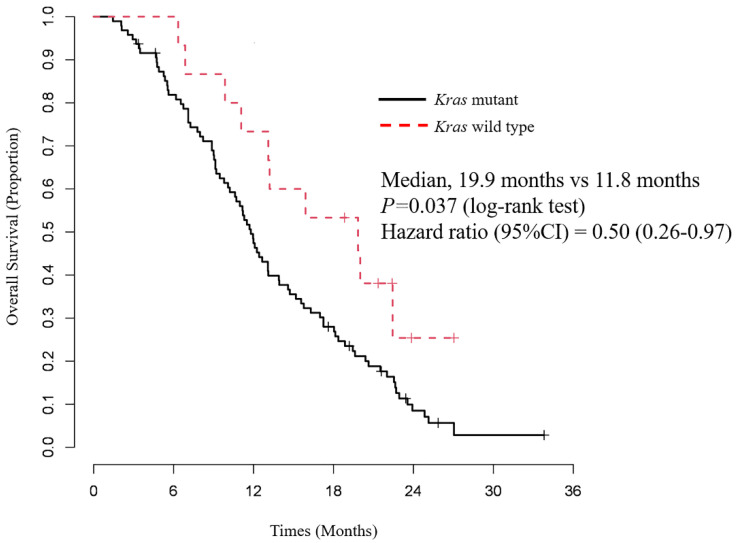
Overall survival according to *Kras* status after first–line chemotherapy with GEM and nab–PTX. The median OS in the wild-type *Kras* and the mutant *Kras* group was 19.9 and 11.8 months, respectively. OS was significantly longer in the wild-type *Kras* group than in the mutant *Kras* group (*p* = 0.037).

**Table 1 cancers-14-00551-t001:** Clinical characteristics of study patients.

	*n* = 110
Sex, male/female	47/73
Patient age, y, mean (range)	67.8 (44–82)
Performance status 0/1	104/6
Lesion size, mm, mean (range)	28.8 (9.8–72)
Location of lesion	
head/body or tail	54/56
Disease status	
local/ metastatic	30/80
CA19-9, mg/mL	
<37/≥37	21/89
*Kras* mutation status	
mutant/wild type	95/15
Amount of extracted DNA, ng/µL, mean (range)	28.2 (1.6–217.0)
Second line chemotherapy *	
absent/present	38/69

* Three patients were still on first-line chemotherapy.

**Table 2 cancers-14-00551-t002:** Characteristics of patients according to the two groups.

	*Kras,* Mutant	*Kras,* Wild Type	*p*-Value
	(*n* = 95)	(*n* = 15)	
Sex, male/female, *n* (%)	55/42 (57.9/42.1)	9/6 (60/40)	1.000
Patient age, y, mean	68.4	64.3	0.102
Performance status 0/1, *n* (%)	88/9 (92.6/7.4)	14/1 (93.3/6.7)	1.000
Lesion size, mm, mean	29.1	27.1	0.524
Location of lesion			
head/body or tail, *n*	42/53 (44.2/55.8)	12/3 (80/20)	0.012
Disease status			
local/metastatic, *n* (%)	24/71 (25.3/74.7)	6/9 (40/60)	0.348
CA19-9, mg/mL			
<37/≥37, *n* (%)	19/78 (20/80)	4/11 (26.7/73.3)	0.480
Amount of extracted DNA, ng/µL, mean	29.4	20.5	0.384
Second line chemotherapy			
absent/present, *n* (%)	32/61 (52.5/47.5)	6/8 (40/60)	0.560

CA19-9, carbohydrate antigen 19-9.

**Table 3 cancers-14-00551-t003:** Response to first-line chemotherapy with GEM and nab-PTX.

t		*Kras,* Mutant	*Kras,* Wild Type	*p*-Value
		(*n* = 95)	(*n* = 15)	
Response, no. (%)	Complete response	0 (0)	0 (0)	
	Partial response	15 (15.8)	3 (20)	
	Stable disease	40 (42.1)	10 (66.7)	
	Progressive disease	40 (42.1)	2 (13.3)	
Rate of objective response *, no. (%)		15 (15.8)	3 (20)	0.701
Rate of disease control **, no. (%)		55 (57.9)	13 (86.7)	0.044

* The rate of objective response was defined as the percentage of patients who had a complete response or partial response. ** The rate of disease control was defined as the percentage of patients who had a complete response, partial response, or stable disease.

**Table 4 cancers-14-00551-t004:** Univariate and multivariate analyses of prognostic factors with progression-free survival.

		Progression-Free Survival			
		Univariate		Multivariate	
		HR (95% CI)	*p*-Value	HR (95% CI)	*p*-Value
Sex	Female/male	0.99 (0.67–1.45)	0.992	0.94 (0.60–1.46)	0.769
Patient age, y	>70/≦70	0.99 (0.67–1.45)	0.946	1.04 (0.68–1.60)	0.842
Performance status	1/0	1.46 (0.64–3.34)	0.368	1.60 (0.65–3.96)	0.309
Lesion size, mm	>20/≦20	1.19 (0.77–1.85)	0.424	1.12 (0.69–1.81)	0.639
Location of lesion	Body or tail /head	1.09 (0.74–1.59)	0.665	1.03 (0.69–1.81)	0.886
Disease status	Local/metastatic	0.83 (0.54–1.28)	0.399	0.98 (0.61–1.57)	0.924
*Kras* status	Wild type/mutant	0.56 (0.31–0.99)	0.049	0.53 (0.28–0.99)	0.045
CA19-9, mg/mL	≥37/<37	0.95 (0.59–1.55)	0.850	0.89 (0.53–1.49)	0.658

CA19-9, carbohydrate antigen 19-9; HR, hazard ratio; 95% CI, 95% confidence interval.

**Table 5 cancers-14-00551-t005:** Univariate and multivariate analyses of prognostic factors associated with overall survival.

		Overall Survival			
		Univariate		Multivariate	
		HR (95% CI)	*p*-Value	HR (95% CI)	*p*-Value
Sex	Female/male	0.87 (0.58–1.31)	0.560	1.10 (0.69–1.77)	0.462
Patient age, y	>70/≦70	0.93 (0.62–1.40)	0.735	0.97 (0.61–1.52)	0.853
Performance status	1/0	1.66 (0.72–3.81)	0.230	1.33 (0.54–3.25)	0.491
Lesion size, mm	>20/≦20	1.26 (0.82–1.83)	0.313	1.10 (0.67–1.81)	0.860
Location of lesion	Body or tail/head	0.82 (0.55–1.83)	0.323	1.16 (0.75–1.79)	0.424
Disease status	Local/metastatic	0.60 (0.37–0.96)	0.026	0.57 (0.36–0.92)	0.048
*Kras* status	Wild type /mutant	0.50 (0.26–0.97)	0.026	0.35 (0.16–0.74)	0.007
CA19-9, mg/mL	≥37/<37	1.02 (0.60–1.74)	0.932	0.62 (0.34–1.10)	0.104
Second line chemotherapy	Present /absent	0.44 (0.29–0.68)	<0.001	0.20 (0.20–0.50)	<0.001

CA19-9, carbohydrate antigen 19-9; HR, hazard ratio; 95% CI, 95% confidence interval.

## Data Availability

The datasets generated during the study will be available from the corresponding author on reasonable request after termination of data collection.

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
