# Peer review of "Kras Gene Analysis Using Liquid-Based Cytology Specimens Predicts Therapeutic Responses and Prognosis in Patients with Pancreatic Cancer"

_cancers, 2022, doi:10.3390/cancers14030551_

Round 1

Reviewer 1 Report

Some minor comments on grammar/readability of this paper: 

Overall the paper is excellently written.  The Simple Summary has many run-on sentences and should be improved with minor edits.  The Discussion could improve readability by removing the word “that” as often as possible. 

A few times in the paper references need to be combined to meet formatting (e.g. Line 55)

Line 123 please spell out nab-paclitaxel and gemcitabine first time used. 

Line 146 likely should read “95 patients with a Kras mutation” or “with a mutant Kras”

Correction:

Line 40, in the reference cited and in more recent Cancer Statistics (ideally cite Cancer Statistics, 2021) pancreas cancer is in fact the THIRD leading cause of cancer death in the United States.  (from 2021 25,270+22,950 = 48,220).

A response is needed for a few questions:

The PFS is only minimally different (1.6 months) where the OS is 8.1 months different.  Why is the focus then on GA as a first line treatment option when it seems maybe subsequent therapies may be more important for OS?

I find the statement “this is the first report suggesting that Kras gene analysis can predict therapeutic response to GA and the prognosis of unresectable PDAC” to be a bit strong.  First, I am not sure it predicts response to GA from the data presented.  I see RECIST response in the Results section, but you do not separate response by KRAS status, so in fact I see no evidence regarding a response difference.  I believe your data demonstrates prognostic information which can be obtained off of EUS FNA. 

The numbers are small but approximately 1/3 of KRAS Mutant patients have local disease and approximately 2/3 of KRAS WT have local disease.  Patients with local disease do better than patients with metastatic disease.  I see the P value is listed as NS, which  I think should be reported and a comment on how this different disease status may affect OS as patients with locally advanced disease lve longer.  I see in table 2 I am told it wasn't significant, and have HR, but I think this needs some more explaiing.

Author Response

Responses to the Reviewer

We thank the editors and reviewers of our manuscript for their helpful comments and suggestions.

Please find our point-by-point responses below. All changes to the manuscript are highlighted in red.

Reviewer: 1

1. 
A few times in the paper references need to be combined to meet formatting (e.g. Line 55)

Authors' reply:

Thank you very much for your suggestion. As you pointed out, we standardized the format of the references in the introduction section (page 3, line 16).

  1. Line 123 please spell out nab-paclitaxel and gemcitabine first time used. 

Authors' reply:

As you point out, we corrected nab-PTX and GEM to nab-paclitaxel and gemcitabine in the Method section (page 6, line 20-21).

  1. Line 146 likely should read “95 patients with a Kras mutation” or “with a mutant Kras”.

Authors' reply:

As you point out, we corrected “with a Kras mutant” to “with a mutant Kras” in the Result section (page 7, line 18-19).

  1. Line 40, in the reference cited and in more recent Cancer Statistics (ideally cite Cancer Statistics, 2021) pancreas cancer is in fact the THIRD leading cause of cancer death in the United States.  (from 2021 25,270+22,950 = 48,220).

Authors' reply:

As you point out, we changed reference [1] and corrected “fourth leading” to “third leading” in the introduction section (page 3, line 10-11).

  1. The PFS is only minimally different (1.6 months) where the OS is 8.1 months different.  Why is the focus then on GA as a first line treatment option when it seems maybe subsequent therapies may be more important for OS?

Authors' reply:

We thank the reviewer for the pointed comments. The association between Kras mutations and prognosis needs to be evaluated in cases with standardized background factors (such as clinical setting and treatment). Therefore, we analyzed Kras mutations in patients with unresectable PDAC who underwent first-line GA therapy. As you mention, the second-line chemotherapy is considered to be very important for OS. However, the second-line chemotherapy in this study was not standardized.

This is one of the limitations of this study, and I added the following paragraph to the limitation section (page 11, line 20-25).

“Third, the presence and type of second-line chemotherapy and disease status (local or metastatic) were not standardized in our study. The second chemotherapy and disease status are considered to be very important for OS. Although Kras status, the presence of second-line chemotherapy and disease status were found to be independent factors for OS, further studies are needed to investigate the relationship between Kras status and OS in patients who have received same chemotherapy regimens and same disease status.”

  1. I find the statement “this is the first report suggesting that Kras gene analysis can predict therapeutic response to GA and the prognosis of unresectable PDAC” to be a bit strong.  First, I am not sure it predicts response to GA from the data presented.  I see RECIST response in the Results section, but you do not separate response by KRAS status, so in fact I see no evidence regarding a response difference.  I believe your data demonstrates prognostic information which can be obtained off of EUS FNA. 

Authors' reply:

Thank you very much for your suggestion. We added analysis of the RECIST response to first-line chemotherapy with GA, according to differences in Kras mutation status. As a result, in the Kras wild-type group, 3 (20%) showed a PR, 10 (66.7%) showed SD, and 2 (13.3%) showed PD, and in the Kras mutant group, 15 patients (15.8%) showed a PR, 40 (42.1%) showed SD, and 40 (42.1%) showed PD. Rate of objective response was observed in 3 patients (20%) in the Kras wild-type group and 15 patients (15.8%) in the Kras mutant group (P=0.701); Table 3). Rate of disease control was observed in 13 patients (86.7%) in the Kras wild-type group and 55 patients (57.9%) in the Kras mutant group (P=0.044); Table 3). There was significant difference in the rate of disease control between two groups. We added this sentence in the result section (Page 6, line 28 and page 7, line 1-8) and Table 3.

We changed “This is the first report suggesting that Kras gene analysis can predict therapeutic responses to GA and the prognosis of unresectable PDAC” to “These findings suggest that Kras gene analysis can predict therapeutic responses to GA, as well as the prognosis of patients with PDAC “ in the discussion section (page 9, line 5-7).

  1. The numbers are small but approximately 1/3 of KRAS Mutant patients have local disease and approximately 2/3 of KRAS WT have local disease.  Patients with local disease do better than patients with metastatic disease.  I see the P value is listed as NS, which  I think should be reported and a comment on how this different disease status may affect OS as patients with locally advanced disease lve longer.  I see in table 2 I am told it wasn't significant, and have HR, but I think this needs some more explaiing.

Authors' reply:

In table 2, % was added. Also, N.S. was changed to a real number in all p-values. The rate of local disease was 25.3% in Kras mutant group and 40% in Kras wild-type group, and there was no significant difference between two groups (P=0.348) (Table 2).

As reviewer indicated, the disease status is considered to be very important for OS. The disease status in this study was not standardized. This is one of the limitations of this study, and I added the following paragraph to the limitation section (page 11, line 20-25)

“Third, the presence and type of second-line chemotherapy and disease status (local or metastatic) were not standardized in our study. The second-line chemotherapy and disease status are considered to be very important for OS. Although Kras status, the presence of second-line chemotherapy and disease status were found to be independent factors for OS, further studies are needed to investigate the relationship between Kras status and OS in patients who have received same chemotherapy regimens and same disease status.”

Reviewer 2 Report

In this manuscript, the authors found that unresectable PDAC patients harboring wild-type Kras had significantly longer progression-free survival and overall survival than those harboring mutant Kras after undergoing first-line gemcitabine and nab-paclitaxel (GA) therapy. They proposed that Kras gene analysis has the potential to predict therapeutic responses to GA and the prognosis of unresectable PDAC. Some revisions should be considered.

  1. Is Kras gene analysis alone sufficient to draw any such conclusion? The authors may consider to compare  whole genome mutational landscape of mutant Kras and wild type Kras groups?
  2. The authors should consider to provide some evidences or to explain possible mechanisms how PDAC harboring wild-type Kras has longer progression-free survival and overall survival than those harboring with mutant Kras after undergoing first-line gemcitabine and nab-paclitaxel (GA) therapy.

Author Response

Responses to the Reviewer

We thank the editors and reviewers of our manuscript for their helpful comments and suggestions.

Please find our point-by-point responses below. All changes to the manuscript are highlighted in red.

Reviewer: 2

  1. Is Kras gene analysis alone sufficient to draw any such conclusion? The authors may consider to compare whole genome mutational landscape of mutant Kras and wild type Kras groups?

Authors' reply:

We agree with the reviewer’s comments. However, we did not analyze the whole genome mutational landscape in this study. In the future, it will be necessary to analyze not only Kras mutation status but also the whole genome mutational landscape to examine the response to GA and OS.

We added the following paragraph to the discussion section (page 10, line 18-22).

These gene mutations may contribute to a better response to GA, although we did not analyze theses gene mutations in this study and the mechanism is unknown. In the future, it will be necessary to analyze not only Kras mutation status but also the whole genome mutational landscape including these mutations to examine the response to GA and OS.”

  1. The authors should consider to provide some evidences or to explain possible mechanisms how PDAC harboring wild-type Kras has longer progression-free survival and overall survival than those harboring with mutant Kras after undergoing first-line gemcitabine and nab-paclitaxel (GA) therapy.

Authors' reply:

Thank you very much for your suggestion. Although, the mechanism for the longer PFS and OS of Kras wild-type patients compared with Kras mutant patients after first-line GA therapy has not been elucidated, we hypothesized that there are three possible reasons.  

We added or revised the following paragraph to the discussion section (Page 9, line 27-28 and page 10, line 11). 

“Although, the mechanism for the longer PFS and OS of Kras wild-type patients compared with Kras mutant patients after first-line GA therapy has not been elucidated, we hypothesized as follows. First, Mutations in Kras cause constitutive activation of RAS and perpetuate downstream signaling pathways involved in cellular proliferation, migration, apoptosis, and cytoskeletal remodeling, independent of growth factor receptor activation [18]. Sustained activation of RAS signaling had been reported as a major cause of gemcitabine resistance[19] and inhibiting Kras signaling suppresses migration and invasion in gemcitabine resistance cell line[20] [21]. Therefore, Kras gene mutation may contribute to acquire resistance and poor response to gemcitabine. Second, Second, Schlitter et al. performed a histological analysis of surgical resection specimens and found that patients with colloid, medullary, or papillary carcinoma survived significantly longer than patients with conventional PDAC (P=0.04) [22]. These findings suggest that Kras wild-type PDACs may be less likely to grow and spread rapidly than Kras-mutant PDACs.”.

Round 2

Reviewer 2 Report

The authors have satisfactorily responded to all my questions.